# CR2PQ: Continuous Relative Rotary Positional Query for Dense Visual Representation Learning

**Shaofeng Zhang**[1]**, Qiang Zhou**[2]**, Sitong Wu**[3]**, Haoru Tan**[4]**, Zhibin Wang**[2]**, Jinfa Huang**[5]**, Junchi Yan**[1*]

[1]Sch. of Computer Science & Sch. of Aitificial Intelligence, Shanghai Jiao Tong University
[2]INF Tech Co., Ltd., [3]CUHK, [4]HKU, [5]Peking University.
{sherrylone, yanjunchi}@sjtu.edu.cn
Code: https://github.com/Sherrylone/PQRoPE

## Abstract

Dense visual representation learning (DRL) shows promise for learning localized information in dense prediction tasks, but struggles with establishing pixel/patch correspondence across different views (cross-contrasting). Existing methods primarily rely on self-contrasting the same view with variations, limiting input variance and hindering downstream performance. This paper delves into the mechanisms of self-contrasting and cross-contrasting, identifying the crux of the issue: transforming discrete positional embeddings to continuous representations. To address the correspondence problem, we propose a Continuous Relative Rotary Positional Query (CR2PQ), enabling patch-level representation learning. Our extensive experiments on standard datasets demonstrate state-of-the-art (SOTA) results. Compared to the previous SOTA method (PQCL), our approach achieves significant improvements on COCO: with 300 epochs of pretraining, CR2PQ obtains **3.4%** mAP$^{bb}$ and **2.1%** mAP$^{mk}$ improvements for detection and segmentation tasks, respectively. Furthermore, CR2PQ exhibits faster convergence, achieving **10.4%** mAP$^{bb}$ and **7.9%** mAP$^{mk}$ improvements over SOTA with just 40 epochs of pretraining.

## 1 Introduction

Self-supervised representation learning (SSL) has been attracting increasing attention in deep learning, whereby the prediction problem is often formulated as a pretext task with pre-training on unlabeled data. SSL methods can mainly be divided into three categories: **1) generative-based** methods (Goodfellow et al., 2014; Donahue & Simonyan, 2019), learning to generate samples in the input space; However, generation can be computationally expensive and may not be necessary for representation learning. **2) contextual-based** methods (Gidaris et al., 2018), designing pretext tasks (*e.g.*,, denoising auto-encoders (Vincent et al., 2008), context autoencoders (Zhang et al., 2016), etc) to pretrain the backbone; **3) contrastive-based** methods (Chen et al., 2020a; He et al., 2020; Chen et al., 2020b; 2021), taking augmented views of the same image as positive pairs and others as negative pairs. Contrastive-based methods have shown great promise in downstream tasks, e.g., image classification/detection (He et al., 2017; Cai & Vasconcelos, 2018) and video classification (Han et al., 2020).

Contrastive learning is a family of instance discrimination-based methods (Chen et al., 2020a; He et al., 2020; Chen et al., 2020b; 2021), which trains the network by distinguishing positive image-level samples from their negative counterparts given query anchors from mini-batches during the learning process. These general (instance-level) contrastive learning (Tian et al., 2020a; Oord et al., 2018) and its many variants (Caron et al., 2021; Zhou et al., 2022; Song & Ermon, 2020) are one of the most popular directions that achieved great success in the past few years and dominate other methods (generative-based and contextual-based) in the field of SSL, especially for linear and finetuning classification tasks.

---
*Junchi Yan is the correspondence author, who is also affiliated with Shanghai Artificial Intelligence Laboratory. Work was partly supported by NSFC (62222607), Shanghai Municipal Science and Technology Major Project (2021SHZDZX0102).

However, these methods (Caron et al., 2021; Chen et al., 2020a; Grill et al., 2020) are still less competitive on dense predictive tasks (Wang et al., 2021; Xie et al., 2021c), *e.g.*, detection (He et al., 2017) and segmentation (Xiao et al., 2018). The main reason is that instance-level contrastive methods aim to learn global-discriminative information, but lack spatial-sensitive information. To address this issue, some dense contrastive learning (DCL) methods with pixel-wise (Xie et al., 2021c; Wang et al., 2021) and patch-wise (Yun et al., 2022) contrastive objectives and frameworks are proposed. However, these methods still have other limitations: **One main shortcoming** of these dense contrastive learning methods is establishing the correspondence among pixels/patches usually requires bilinear interpolation, which is complex and heavily sensitive to random crop augmentation (in an extreme case, if two views have no intersection parts, there are no correspondence relation). One simple way to avoid mining the correspondence is inputting the same view (masked and unmasked versions) twice (Zhou et al., 2022). However, the variance of the inputs (masked and unmasked views) is much lower than inputting two different views, where the variance of two views has been proven to be the key to success in contrastive learning (Wang et al., 2022a; Tian et al., 2020b).

To address the correspondence problem, we propose a continuous rotary positional-query-based paradigm. Specifically, we first randomly crop two views from the image, and use the relative coordinate system and rotary positional embedding to represent the relative positions between the two views. Then, we take the rotary relative positional embedding as a query and reconstruct the latent representations (or RGB value) of one view from the other view. Through the relative coordinate system rotary positional query, our CR2PQ significantly simplifies the previous DCL paradigm (usually using GCN or other sub-network to learn the correspondence information), making the dense contrastive learning correspondence-free. Our main contributions are:

**i) Correspondence-free DCL via relative coordinate system and continuous rotary positional embedding.** Instead of simply employing RoPE to the vision domain (1D to 2D), we further transform the discrete positional embedding to continuous, taking the relative coordinates as input, which makes our CR2PQ correspondence-free. Besides, due to the simplicity of CR2PQ, our methods can be integrated into several popular representation learning paradigms, *e.g.*, MIM-based, CL-based, and Distillation-based methods (See Table 4).

**ii) Positional-aware query module.** Instead of directly using the vanilla self attention (Vaswani et al., 2017) (which would hurt the downstream performance and bring more computational cost due to the extra query tokens, as the downstream task only inputs the raw tokens), we propose positional-aware cross attention module in the final block between query tokens and patches embeddings to learn semantic information of query tokens, and the cross attention only incurs a few extra parameters.

**iii) New SOTA performance.** We conduct exhaustive experiments on classification, detection, and segmentation tasks, where our CR2PQ achieves new SOTA results. In particular, CR2PQ outperforms previous DCL methods with a large range, especially on dense prediction tasks. Specifically, it outperforms previous SOTA by **3.4%**, and **2.1%** points for detection and segmentation on the MSCOCO dataset. CR2PQ surpasses previous SOTA PQCL **1.9%** points on ADE20K semantic segmentation. We further conduct experiments to show its convergence speed, and we surprisingly find with few epochs (*e.g.*, 40) pretraining, CR2PQ outperforms previous SOTA DCL methods (Zhang et al., 2023a) **10.4%** and **7.9%** scores on detection and segmentation on the MSCOCO dataset.

## 2 RELATED WORKS

**Instance contrastive learning.** Instance self-supervised learning aims to extract informative image representations by leveraging unlabeled data. It achieves this by applying contrastive objectives, which essentially bring similar representations closer while pushing dissimilar ones further apart. A key challenge in contrastive learning is handling negative examples, which are data points used to differentiate the representation of the target image. Memory-based methods like MoCo (He et al., 2020; Chen et al., 2020b) store negative examples in a memory bank for comparisons. SimCLR (Chen et al., 2020a), on the other hand, treats all other data points within the same batch as negative examples. To avoid the need for explicit negative examples, BYOL (Grill et al., 2020) and SimSiam (Chen & He, 2021) employ separate encoder and predictor networks with a stop-gradient mechanism. Barlow Twins (Zbontar et al., 2021), ZeroCL (Zhang et al., 2021), and VICReg (Bardes et al., 2022) achieve similar results by focusing on decorrelating features at different levels. Building upon ZeroCL (Zhang et al., 2021), ARB (Zhang et al., 2022) proposes aligning representations with an orthogonal base for computational efficiency. Inspired by the success of Vision Transformers (ViTs), recent work

like MoCo v3 (He et al., 2020) and DINO (Caron et al., 2021) are replacing convolutional neural networks (CNNs) with ViT backbones.

**Dense contrastive self-supervised learning.** In contrast to general contrastive learning, dense contrastive learning (Wang et al., 2021; Yang et al., 2021; Ziegler & Asano, 2022; Ge et al., 2021) methods aim to learn spatial-sensitive information to provide a pretrained model to dense predictive tasks (*e.g.*, object detection, instance segmentation and semantic segmentation). DenseCL (Wang et al., 2021) exploits the correspondence by sorting the similarities of pixels in the deep feature map, while PixPro (Xie et al., 2021c) utilizes the augmentation wrapper to get the spatial correspondence of the pixel intersection between two views. Furthermore, Detco (Xie et al., 2021a) tries to improve the performance of general contrastive learning approaches by augmenting multiple global and local views simultaneously. Inspired by PixPro, Resim (Xiao et al., 2021) uses RoI Pooling (Jiang et al., 2018) to extract a feature vector from the associated feature map region for both views. On the basis of DenseCL, SetSim (Wang et al., 2022b) employs a threshold selection to filter out noisy backgrounds. With the development of ViT in SSL (Dosovitskiy et al., 2020), SelfPatch (Yun et al., 2022) treats the spatial neighbors of the patch as positive examples for learning more semantically meaningful relations among patches. On the basis of DINO (Caron et al., 2021), ADCLR (Zhang et al., 2023b) proposes patch-level contrasting via unmasked query tokens and cross-attention mechanism to avoid mining spatial correspondence. To further increase the variance of patch-level contrasting, PQCL (Zhang et al., 2023a) proposes to replace the unmasked query tokens with the relative positional embeddings, which further increase the difficulty of DCL, resulting in better performance in dense predictive tasks.

**Masked Image Modeling (MIM).** Masked Image Modeling (MIM) is a self-supervised learning technique where the model reconstructs masked portions of an image to learn meaningful representations. MIM methods can be categorized based on the information they reconstruct: (a) raw pixel values, (b) auxiliary features, and (c) masked patch embeddings. SimMIM (Xie et al., 2022) and MAE (He et al., 2021) (a) reconstruct the raw pixel values from masked or partially observed patches. MaskFeat (Wei et al., 2021) (b) incorporates HOG features as supervisory signals for richer semantics. CIM (Fang et al., 2022) (a) enhances robustness by adding perturbations to raw images. iBOT (Zhou et al., 2022) (c) is the first to predict token embeddings, but using the same view might influence downstream tasks. SIM (Tao et al., 2022) (c) addresses this by predicting masked patch embeddings from a different view, but introduces higher computational cost. MIM research offers a promising approach for learning powerful image representations for various computer vision tasks.

**RoPE in vision domain.** Relative Positional Encoding (RoPE) (Su et al., 2024) has emerged as a promising technique to improve vision transformers. Pioneering studies introduced RoPE to ViT-based architectures, with the Hybrid X-former (Jeevan & Sethi, 2022) applying 1D RoPE. While this approach demonstrates the potential of RoPE, its limitations in capturing spatial relationships hinder performance on complex vision tasks like classification, detection, and segmentation. Additionally, evaluations using small datasets might not accurately reflect the effectiveness of RoPE on larger and more intricate datasets. Recent studies exploring 2D RoPE, such as EVA-02 (Fang et al., 2023) and Unified-IO 2 (Lu et al., 2023), focused on language-related tasks or new model architectures. This work aims to bridge this gap by investigating the effectiveness of 2D RoPE in improving the performance of basic architectures on challenging vision tasks.

## 3 METHODOLOGY

### 3.1 PRELIMINARIES

**APE and Vision Transformers.** Denote an image by $\mathbf{x} \in \mathbb{R}^{C \times H \times W}$, where $H \times W$ is the resolution of the image and $C$ is the number of channels. Plain ViT (Dosovitskiy et al., 2020) treats the image $\mathbf{x}$ as a sequence composed of non-overlapping patches $\{\mathbf{x}^{(i)} \in \mathbb{R}^{CP^2}\}_{i=1}^{N}$, where each patch has a fixed $P \times P$ resolution. Then, the patches are linearly transformed to $D$-dimensional patch embeddings $\mathbf{z}^{(i)} = \mathbf{E}\mathbf{x}^{(i)} + \mathbf{P}_{pos}^{i} \in \mathbb{R}^{D}$, where $\mathbf{E} \in \mathbb{R}^{D \times CP^2}$ is the linear projection and $\mathbf{P}_{pos}^{i} \in \mathbb{R}^{D}$ is the positional embedding for the $i$-th patch. A $[CLS]$ token $\mathbf{z}^{([CLS])} \in \mathbb{R}^{D}$ is subsequently prepended to the patch sequence to extract global information, so the resulting input sequence is represented as $\mathbf{z} = [\mathbf{z}^{([CLS])}, \mathbf{z}^{(1)}, \mathbf{z}^{(2)}, \cdots, \mathbf{z}^{(N)}]$. Then, ViT uses a Transformer encoder (Vaswani et al., 2017) to generate both image-level ($[CLS]$ token) and patch-level (other tokens). In line with SelfPatch (Yun

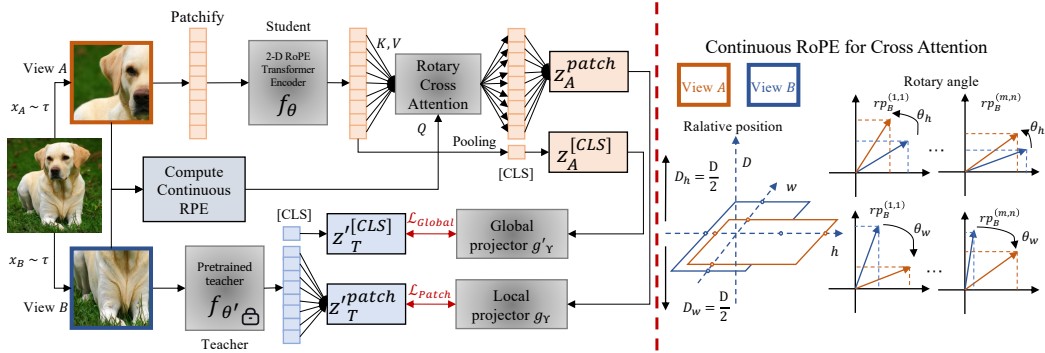

Figure 1: framework of the proposed CR2PQ. We first crop the image twice along with augmentations to generate view $A$ and view $B$. Then, we compute the continuous relative position of the two views (illustrated in the right figure). For view $A$ and view $B$, we feed them into the transformer encoder and the pre-trained teacher network, respectively. Then, we use the pre-calculated rotary positional embedding to extract patch-level embeddings of view $B$ through the rotary cross-attention block. Finally, the patch-level and global alignment objectives are added to learn dense representations.

et al., 2022), we use $f_\theta$ to denote the whole process of a ViT parameterized by $\theta$:

$$f_\theta(\mathbf{x}) = f_\theta\left(\left[\mathbf{z}^{([CLS])}, \mathbf{z}^{(1)}, \mathbf{z}^{(2)}, \cdots, \mathbf{z}^{(N)}\right]\right) = \left[\mathbf{o}^{([CLS])}, \mathbf{o}^{(1)}, \mathbf{o}^{(2)}, \cdots, \mathbf{o}^{(N)}\right] \quad (1)$$

where $\mathbf{o}^{[CLS]}$ and $\mathbf{o}^{(i)}$ are the representations of the whole image and $i$-th patch, respectively.

**Rotary positional embedding in 1-D.** In Transformer models, self-attention plays a crucial role in capturing relationships between words in a sentence. However, self-attention mechanisms lack inherent knowledge of word order. Positional encoding techniques address this by injecting positional information into the model. Traditional discrete relative positional encoding (RPE) methods (take the discrete patch index as input, and return the positional embeddings), like relative positional bias (RPB), are often limited by the interaction with attention weights, which causes limited utilization of relative position. Thus, RoFormer (Su et al., 2024) proposes a novel relative position embedding method: Rotary Position Embedding (RoPE). RoPE directly incorporates relative position information into the attention computation by applying trigonometric functions (sine and cosine) to key and query vectors. This allows RoPE to interact effectively with attention weights, leading to a comprehensive understanding of word order within the self-attention process. Specifically, RoPE introduces the multiplication of Euler's formula ($e^{i\theta}$) to key and query vectors as relative position embedding, *i.e.*, when $n$, $m$-th query and keys in $\mathbf{q}_n, \mathbf{k}_m \in \mathbb{R}^{1 \times D_{head}}$, RoPE is applied as follows:

$$\mathbf{q}'_n = \mathbf{q}_n e^{in\theta}, \quad \mathbf{k}'_m = \mathbf{k}_m e^{im\theta}, \quad Attn_{n,m} = Re[\mathbf{q}'_n \mathbf{k}'^*_m] = Re[\mathbf{q}_n \mathbf{k}^*_m e^{i(n-m)\theta}] \quad (2)$$

where $Re[\cdot]$ denotes real part of complex number and $^*$ means complex conjugates.

## 3.2 THE PROPOSED CR2PQ

**View Generation.** Given the image $\mathbf{x} \in \mathbb{R}^{H \times W \times 3}$ in training set, we first randomly crop the image twice to generate two views $\mathbf{x}_A$ and $\mathbf{x}_B$. Then, we resize the two views to $H \times W \times 3$. Note that we randomly crop two views, therefore, the positional relationship of the two views might be either containing, overlapping, or non-overlapping. We record the absolute position $\mathbf{p}_A$ of $\mathbf{x}_A$ and $\mathbf{p}_B$ of $\mathbf{x}_B$, respectively. Specifically, $\mathbf{p}_A$ and $\mathbf{p}_B$ are composed of $\mathbf{p}_A = \{p_{Ai}, p_{Aj}, p_{Ah}, p_{Aw}\}$ (top location, left location, height and width) and $\mathbf{p}_B = \{p_{Bi}, p_{Bj}, p_{Bh}, p_{Bw}\}$, respectively.

**Computing Relative Positional Coordinates.** The core idea of our CR2PQ is to use the relative coordinate system and continuous rotary positional embeddings to represent the positional relationship between the two views. For simplicity, we use view $A$ to reconstruct view $B$ as an example. Specifically, Suppose the grid size of the feature map of the view $A$ is $K_A$. Then, the patch index matrix of view $A$ can be written as:

$$\mathbf{rp}_A = \begin{pmatrix} (0,0) & (0,1) & (0,2) & \cdots & (0, K_A^W - 1) \\ (1,0) & (1,1) & (1,2) & \cdots & (1, K_A^W - 1) \\ (2,0) & (2,1) & (2,2) & \cdots & (2, K_A^W - 1) \\ \cdots & \cdots & \cdots & \cdots & \cdots \\ (K_A^H - 1, 0) & (K_A^H - 1, 1) & (K_A^H - 1, 2) & \cdots & (K_A^H - 1, K_A^W - 1) \end{pmatrix} \quad (3)$$

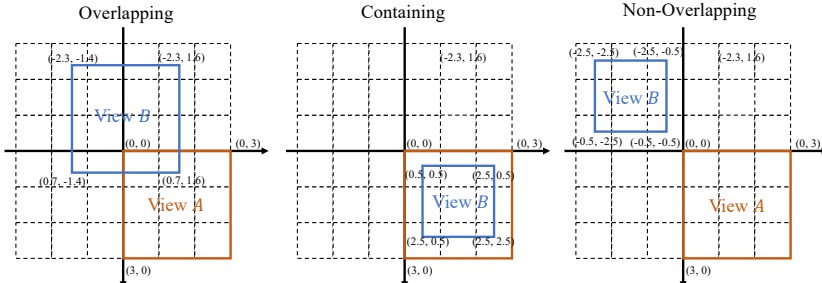

Figure 2: Illustration of three different scenarios (overlapping, containing, and non-overlapping). The proposed relative coordinate system can easily represent the relative coordinate matrix $\mathbf{rp}_B$ in both three scenarios. We always use the top-left corner of the view $A$ as the origin point and set the interval of the feature map of view $A$ as 1.

where $K_A^H = H/ps_A$, and $K_A^W = W/ps_A$, and $ps_A$ is the patch size when inputting the view $A$ to the model. Observed through Eq. 3, we can recognize the patch index matrix $\mathbf{rp}_A$ as a 2-D coordinates system, where the top-left element is the origin point and the scale of the coordinates system is 1. The detailed illustration can be found in Fig. 2. Since we set each grid size of the feature map of the anchor view as 1, then the relative positional indices (relative coordinates) of the patches of the view $B$ can be written as:

$$\mathbf{rp}_B^{m,n}(ps_A, ps_B) = \left( \underbrace{\frac{K_A^H \cdot (p_{Bi} - p_{Ai})}{p_{Ah}} + \frac{p_{bh} \cdot K_A^H \cdot (m-1)}{K_B^H \cdot p_{Ah}}}_{\text{Row}}, \right.$$
$$\left. \underbrace{\frac{K_A^W \cdot (p_{Bj} - p_{Aj})}{p_{Aw}} + \frac{p_{Bw} \cdot K_A^W \cdot (n-1)}{K_B^W \cdot p_{Aw}}}_{\text{Column}} \right) \tag{4}$$

where $(\mathbf{rp}_B^m, \mathbf{rp}_B^n)$ is the $m$-th row $n$-th column element in the coordinate matrix $\mathbf{rp}_B$. $ps_B$ is the patch size when inputting the view $B$ to the model, and $K_B^H = H/ps_B$, $K_B^W = W/ps_B$. Note that $ps_A$ and $ps_B$ are dependent on the patch size of the predefined backbone and teacher networks.

**Continuous Relative Rotary Positional Embeddings.** Given the coordinate indices matrix $\mathbf{rp}_A$ of view $A$, suppose the hidden dimensions of each head of the backbone is $D_{head}$, then we first convert query and key to $\mathbb{C}^{1 \times (D_{head}/4)}$ by considering the $2t$-th dim as real part and $2t+1$-th dim as imaginary part. For each channel, following RoPE (Su et al., 2024), we utilize multiple frequencies $\theta_t$ in the key and query, *i.e.*, $\theta_t = 10000^{-t/(D_{head}/2)}$, where $t \in \{0, 1, \cdots, D_{head}/4\}$. Then, for the $m$-th row and $n$-th column location in the positional matrix in Eq. 3 of view $A$, the rotation matrix $\mathbf{R}_{rot} \in \mathbb{C}^{(K_A^H K_A^W) \times D_{head}/4}$ can be written as:

$$\begin{aligned} \mathbf{R}_{rot}^{(m,n,2t)} = e^{i\theta_t m}, \quad \mathbf{R}_{rot}^{(m,n,2t+1)} = e^{i\theta_t m}, \quad 0 \le t < D_{head}/2 \\ \mathbf{R}_{rot}^{(m,n,2t)} = e^{i\theta_t n}, \quad \mathbf{R}_{rot}^{(m,n,2t+1)} = e^{i\theta_t n}, \quad D_{head}/2 \le t < D_{head} \end{aligned} \tag{5}$$

Then, for view $B$, we can replace the $m$, $n$ in Eq. 5 with $\frac{K_A^H \cdot (p_{Bi} - p_{Ai})}{p_{Ah}} + \frac{p_{bh} \cdot K_A^H \cdot (m-1)}{K_B^H \cdot p_{Ah}}$ and $\frac{K_A^W \cdot (p_{Bj} - p_{Aj})}{p_{Aw}} + \frac{p_{Bw} \cdot K_A^W \cdot (n-1)}{K_B^W \cdot p_{Aw}}$, respectively. Then, the relative rotary matrix becomes to:

$$\mathbf{R}_{rela}^{(m,n)} = \begin{cases} e^{i\theta_t \left[ \frac{K_A^H \cdot (p_{Bi} - p_{Ai})}{p_{Ah}} + \frac{p_{bh} \cdot K_A^H \cdot (m-1)}{K_B^H \cdot p_{Ah}} \right]}, & 0 \le t < D_{head}/2 \\ e^{i\theta_t \left[ \frac{K_A^W \cdot (p_{Bj} - p_{Aj})}{p_{Aw}} + \frac{p_{Bw} \cdot K_A^W \cdot (n-1)}{K_B^W \cdot p_{Aw}} \right]}, & D_{head}/2 \le t < D_{head} \end{cases} \tag{6}$$

where $2t$-th dims are the real part and $2t+1$-th dims are the imaginary part.

**Framework and Objectives.** Given the cropped views $\mathbf{x}_A$, $\mathbf{x}_B$, we first calculate the relative positions $\mathbf{rp}_A$ and $\mathbf{rp}_B$. Then, we calculate the rotary matrix $\mathbf{R}_{rot}$ and relative rotary $\mathbf{R}_{rela}$ via Eq. 5 and Eq. 6, respectively. Then, we feed the $\mathbf{x}_A$ to the encoder network, which is composed of a

patchify layer and several transformer blocks. For each block, we replace the self-attention with:

$$\text{Attn}(\mathbf{Q}, \mathbf{K}, \mathbf{V})^{(m,n)} = \sum_{h=0}^{K_A^{H-1}} \sum_{w=0}^{K_A^{W-1}} \frac{\exp\left((\mathbf{R}_{rot}^{(m,n)}\mathbf{q}^{(m,n)})^\top (\mathbf{R}_{rot}^{(h,w)}\mathbf{k}^{(h,w)})\cdot\right)\mathbf{v}^{(h,w)}}{\sum_{h=0}^{K_A^{H-1}} \sum_{w=0}^{K_A^{W-1}} \exp\left((\mathbf{R}_{rot}^{(m,n)}\mathbf{q}^{(m,n)})^\top (\mathbf{R}_{rot}^{(h,w)}\mathbf{k}^{(h,w)})\right)} \tag{7}$$

where $\mathbf{q}^{(m,n)} = \mathbf{h}^{(m,n)}\mathbf{W}_Q$, $\mathbf{k}^{(m,n)} = \mathbf{h}^{(m,n)}\mathbf{W}_K$, and $\mathbf{v}^{(m,n)} = \mathbf{h}^{(m,n)}\mathbf{W}_V$, where $\mathbf{h}^{(m,n)}$ is the hidden representation of the patch located in $m$-th row and $n$-th column. $\mathbf{W}_Q, \mathbf{W}_K, \mathbf{W}_V$ are learnable parameters. Given the parameterized backbone by $f_\theta$, we can calculate the representation of the view $A$ by $\mathbf{h}_A = f_\theta(\mathbf{x}_A, \mathbf{R}_{rot})$. Then, the position-aware cross attention becomes:

$$\mathbf{h}^{\mathbf{rp}_B^{m,n}} = \sum_{h=0}^{K_A^{H-1}} \sum_{w=0}^{K_A^{W-1}} \frac{\exp\left((\mathbf{R}_{rela}^{(m,n)}\Phi(\mathbf{rp}_B^{(m,n)}))^\top (\mathbf{R}_{rot}^{(h,w)}\mathbf{k}^{(h,w)})\cdot\right)\mathbf{v}^{(h,w)}}{\sum_{h=0}^{K_A^{H-1}} \sum_{w=0}^{K_A^{W-1}} \exp\left((\mathbf{R}_{rela}^{(m,n)}\Phi(\mathbf{rp}_B^{(m,n)}))^\top (\mathbf{R}_{rot}^{(h,w)}\mathbf{k}^{(h,w)})\right)} \tag{8}$$

where $\mathbf{R}_{rela}^{(m,n)}$ is given by Eq. 6, and $\mathbf{rp}_B^{(m,n)}$ is the $m$-th row $n$-th column elements of the relative coordinate matrix $\mathbf{rp}_B$, as derived by Eq. 4. $\Phi(\mathbf{rp}_B^{(m,n)}) = (\mathbf{q}^{(m,n)} + \mathbf{h}_{mask})\mathbf{W}_Q$, where $\mathbf{h}_{mask}$ is the learnable masked placeholder, which is also commonly used in other methods (Zhou et al., 2022; Xie et al., 2022), and $\mathbf{W}_Q$ is the learnable parameters. After obtaining the $\mathbf{h}^{\mathbf{rp}_B}$, we feed the $\mathbf{h}^{\mathbf{rp}_B}$ to the light-weight decoder $g_\gamma$ to obtain $\mathbf{z}_B$, and predict the embedding or RGB pixel values of the view $B$. Specifically, for distillation methods, we adopt a pretrained encoder $f'_\theta(\cdot)$, and extract the hidden representations by $\mathbf{z}'_B = f'_\theta(\mathbf{x}_B)$, where $\mathbf{z} \in \mathbb{R}^{K_B^H \times K_B^H \times D_{tea}}$, where $D_{tea}$ is the output dimension of the teacher model. Finally, we compute the patch-level objective by:

$$\mathcal{L}_{Patch} = \frac{1}{K_B^H K_B^W} \sum_{h=1}^{K_B^H} \sum_{w=1}^{K_B^W} \left\| \mathbf{z}_B^{(h,w)} - \mathbf{z}'_B^{(h,w)} \right\|_p^p, \tag{9}$$

Meanwhile, we add another global objective term to learn global-discriminative information by:

$$\mathcal{L}_{Global} = \left\| g'_\gamma \left( \frac{1}{K_B^H K_B^W} \sum_{h=1}^{K_B^H} \sum_{w=1}^{K_B^W} \mathbf{h}_T^{(h,w)} \right) - \frac{1}{K_B^H K_B^W} \sum_{h=1}^{K_B^H} \sum_{w=1}^{K_B^W} \mathbf{z}'_T^{(h,w)} \right\|_p^p, \tag{10}$$

where $g'_\gamma$ is the global projector composed of two linear layers and an activation function. Finally, we adopt a hyper-parameter to balance the global and local objectives $\mathcal{L}_{CR2PQ} = \mathcal{L}_{Patch} + \lambda \cdot \mathcal{L}_{Global}$.

## 4 EXPERIMENTS

**Datasets.** We conduct self-supervised pre-training on the ImageNet-1K (Deng et al., 2009) training set with 1,000 classes, as used in SSL for both MIM (He et al., 2021) and contrastive learning (Chen et al., 2020a). We also transfer the encoder pre-trained by CR2PQ on MS-COCO (Lin et al., 2014) and ADE20K (Zhou et al., 2017) datasets.

**Pre-training hyper-parameters.** In line with CAE (Chen et al., 2022), we train with Adamw (Loshchilov & Hutter, 2018) and a batch size of 2048, distributed over 32 GPUs using ViT-S/16 (batch size per GPU is 64). For ViT-B, the learning rate is linearly ramped up during the first 40 epochs to its base value determined with the following linear scaling rule (Chen et al., 2020a): $blr$ = 1.5e-4, $BatchSize$=2048, and $lr = blr * BatchSize/256$. For ViT-S, we set $blr$ as 1.75e-4. After warmup, we decay the learning rate with a cosine schedule (Loshchilov & Hutter, 2016). We follow the data augmentations of BYOL (Grill et al., 2020) (color jittering, Gaussian blur, and solarization) with a bicubic interpolation to adapt the position embeddings to the scales.

**Platform.** The experiments are performed on a workstation with 32 V100 GPUs by default (if not otherwise specified).

### 4.1 MAIN RESULTS

**COCO object detection and segmentation. Setups.** We evaluate pre-trained models on the COCO object detection and instance segmentation tasks (Lin et al., 2014). We evaluate our model under two

Table 1: **Accuracy on MS-COCO**. Mask R-CNN (He et al., 2017) and Cascade R-CNN (Cai & Vasconcelos, 2018) are adopted and trained with the 1x schedule. All the results are obtained by using our same finetune protocol for fair comparisons. Epoch refers to the number of pretraining.

| Method | Backbone | Framwork | #Epochs | #Param. | #Views. | Object Detection | | | Instance Segmentation | | |
|---|---|---|---|---|---|---|---|---|---|---|---|
| | | | | | | $AP^{bb}$ | $AP^{bb}_{50}$ | $AP^{bb}_{75}$ | $AP^{mk}$ | $AP^{mk}_{50}$ | $AP^{mk}_{75}$ |
| Moco-V2 (Chen et al., 2020b) | ResNet-50 | | 200 | 23M | $2 \times 224^2$ | 38.9 | 59.2 | 42.4 | 35.5 | 56.2 | 37.8 |
| SwAV (Caron et al., 2020) | ResNet-50 | | 200 | 23M | $2 \times 224^2$ | 38.5 | 60.4 | 41.4 | 35.4 | 57.0 | 37.7 |
| DenseCL (Wang et al., 2021) | ResNet-50 | | 200 | 23M | $2 \times 224^2$ | 40.3 | 59.9 | 44.3 | 36.4 | 57.0 | 39.2 |
| ReSim (Xiao et al., 2021) | ResNet-50 | | 200 | 23M | $2 \times 224^2$ | 40.3 | 60.6 | 44.2 | 36.4 | 57.5 | 38.9 |
| DetCo (Xie et al., 2021a) | ResNet-50 | | 200 | 23M | $2 \times 224^2$ | 40.1 | 61.0 | 43.9 | 36.4 | 58.0 | 38.9 |
| Moco V3 (Chen et al., 2021) | ViT-S/16 | Mask RCNN | 300 | 23M | $2 \times 224^2$ | 39.8 | 62.6 | 43.1 | 37.1 | 59.6 | 39.2 |
| MoBY (Xie et al., 2021b) | ViT-S/16 | | 300 | 22M | $2 \times 224^2$ | 41.1 | 63.7 | 44.8 | 37.3 | 60.3 | 39.8 |
| DINO (Caron et al., 2021) | ViT-S/16 | | 300 | 22M | $2 \times 224^2$ | 40.8 | 63.4 | 44.2 | 37.3 | 59.9 | 39.5 |
| SelfPatch (Yun et al., 2022) | ViT-S/16 | | 200 | 22M | $2 \times 224^2$ | 42.1 | 64.9 | 46.1 | 38.5 | 61.3 | 40.8 |
| iBOT (Zhou et al., 2022) | ViT-S/16 | | 200 | 22M | $2 \times 224^2$ | 42.6 | 65.7 | 47.0 | 39.0 | 61.7 | 41.3 |
| PQCL (Zhang et al., 2023a) | ViT-S/16 | | 200 | 22M | $2 \times 224^2$ | 43.1 | 66.0 | 47.4 | 39.3 | 62.2 | 41.6 |
| PQCL (Zhang et al., 2023a) | ViT-S/16 | | 300 | 22M | $2 \times 224^2$ | 44.0 | 66.7 | 48.1 | 39.7 | 63.1 | 42.2 |
| CR2PQ (Ours) | ViT-S/16 | | 200 | 22M | $2 \times 224^2$ | 45.0 | 67.4 | 49.0 | 39.9 | 63.4 | 42.7 |
| CR2PQ (Ours) | ViT-S/16 | | 300 | 22M | $2 \times 224^2$ | **47.4** | **69.3** | **52.0** | **41.8** | **65.9** | **44.7** |
| DINO (Caron et al., 2021) | ViT-S/16 | | 300 | 22M | $2 \times 224^2$ | 45.2 | 64.9 | 47.8 | 38.9 | 61.2 | 41.7 |
| SelfPatch (Yun et al., 2022) | ViT-S/16 | | 300 | 22M | $2 \times 224^2$ | 46.6 | 65.7 | 48.8 | 39.5 | 62.0 | 42.6 |
| DINO (Caron et al., 2021) | ViT-S/16 | | 800 | 22M | $2 \times 224^2 + 10 \times 96^2$ | 46.8 | 66.7 | 50.3 | 40.6 | 63.7 | 43.2 |
| iBOT (Zhou et al., 2022) | ViT-S/16 | Cascade RCNN | 300 | 22M | $2 \times 224^2$ | 45.4 | 65.1 | 49.0 | 39.6 | 62.1 | 41.7 |
| iBOT (Zhou et al., 2022) | ViT-S/16 | | 800 | 22M | $2 \times 224^2 + 10 \times 96^2$ | 49.4 | 68.7 | 53.3 | 42.6 | 65.6 | 45.8 |
| PQCL (Zhang et al., 2023a) | ViT-S/16 | | 200 | 22M | $2 \times 224^2$ | 46.2 | 65.5 | 49.8 | 39.9 | 62.3 | 42.6 |
| PQCL (Zhang et al., 2023a) | ViT-S/16 | | 300 | 22M | $2 \times 224^2$ | 47.7 | 67.0 | 51.3 | 41.1 | 64.0 | 44.2 |
| RoPE (Su et al., 2024) ( Scratch) | ViT-S/16 | | 0 | 22M | $2 \times 224^2$ | 31.0 | 47.1 | 33.4 | 27.7 | 44.9 | 29.2 |
| CR2PQ (Ours) | ViT-S/16 | | 100 | 22M | $2 \times 224^2$ | 49.4 | 68.1 | 53.5 | 42.7 | 65.4 | 46.1 |
| CR2PQ (Ours) | ViT-S/16 | | 200 | 22M | $2 \times 224^2$ | 50.3 | 69.2 | 54.4 | 43.2 | 66.5 | 46.5 |
| CR2PQ (Ours) | ViT-S/16 | | 300 | 22M | $2 \times 224^2$ | **50.5** | **69.5** | **54.7** | **43.4** | **66.7** | **46.8** |

popular frameworks Mask-RCNN (He et al., 2017) and Cascade R-CNN (Cai & Vasconcelos, 2018) with the standard 1x schedule (12 epochs). In line with previous methods (Zhou et al., 2022; Zhang et al., 2023a), we adopt AdamW (Loshchilov & Hutter, 2018) optimizer and set the learning rate as 3e-4 with weight decay 0.05. **Evaluation.** MS COCO (Lin et al., 2014) is a large-scale object detection, segmentation, and captioning dataset: in particular, train 2017 and val 2017 splits contain $118K$ and $5K$ images, respectively. We follow the basic configuration of mmdetection (Chen et al., 2019) for fine-tuning Mask R-CNN (He et al., 2017) with FPN (Lin et al., 2017) under the standard 1x schedule. **Results.** Table 1 shows the proposed CR2PQ can consistently outperform previous SOTA (Zhang et al., 2023a) in both object detection and instance segmentation tasks. We evaluate CR2PQ with both 200 and 300 epochs pretraining. When applying the pre-trained model to Cascaded-RCNN, for 200 epochs pretraining without local views (Caron et al., 2020) and query views (Zhang et al., 2023a), CR2PQ surpasses PQCL (Zhang et al., 2023a) with 100 epochs pretraining and one extra query view pretraining **1.7%** point mAP$^{bb}$ and **1.6%** point mAP$^{mk}$, respectively. Besides, our CR2PQ with 300 epochs pretraining can outperform iBOT (800 epochs pretraining with 10 extra local views) by **1.1%** point mAP$^{bb}$ and **0.8%** point mAP$^{mk}$. When adopting the Mask-RCNN framework, under 200 epochs pretraining, our model CR2PQ outperforms PQCL by **1.9%** point mAP$^{bb}$ and **0.6%** point mAP$^{mk}$, respectively. With 300 epochs pretraining, compared with previous SOTA PQCL (Zhang et al., 2023a), our CR2PQ achieves **3.4%** mAP$^{bb}$ and **2.1%** mAP$^{mk}$ improvements on detection and segmentation tasks, respectively. We also evaluate the detection and segmentation performance without pretraining, *i.e.*, directly using 2D RoPE (Su et al., 2024), where the 2D RoPE obtains much worse results than CR2PQ.

**ADE20K semantic segmentation. Setup.** We evaluate semantic segmentation performances of pre-trained models on ADE20K (Zhou et al., 2017), which contains 150 fine-grained semantic categories and $25K$ training data. We finetune the pre-trained models on Semantic FPN (Lin et al., 2017) and UperNet (Xiao et al., 2018) with $40K$ and $160K$ iteration, respectively. Following the previous methods SelfPatch (Yun et al., 2022) and PQCL (Zhang et al., 2023a), we report three metrics: (a) mean intersection of union (mIoU) averaged over all semantic categories, (b) all pixel accuracy (aAcc), and (c) mean class accuracy (mAcc). **Evaluation.** ADE20K (Zhou et al., 2017) is a semantic segmentation benchmark containing 150 fine-grained semantic categories and $25K$ images. We follow all the configurations of mmsegmentation (Contributors, 2020) for fine-tuning Semantic FPN (Lin et al., 2017) with $40K$ iterations and an input resolution of 512×512. We also perform large-scale fine-tuning experiments using UPerNet (Xiao et al., 2018) with $160K$ iterations and an input resolution of 512×512. **Results.** As shown in Table 2, CR2PQ can outperform previous all methods under the same setting. Besides, we further evaluate DINO (Caron et al., 2021) with 10 local views and 800 epochs pretraining (checkpoint is downloaded in their official repository), where CR2PQ gets **2.6%** point improvements with only 200 epochs pretraining and only two global views. We guess the big improvements are because CR2PQ is a patch-level distillation method, which makes our method more spatial-sensitive, resulting the higher performance in dense prediction tasks.

Table 2: **ADE20K semantic segmentation** performances of the recent self-supervised approaches pre-trained on ImageNet. The metrics mIoU, aAcc, and mAcc denote the mean intersection of union, all pixel accuracy, and mean class accuracy, respectively.

| Method | Arch | Backbone | #Iter | #Epochs | #Params | #Views | mIoU | aAcc | mAcc |
|---|---|---|---|---|---|---|---|---|---|
| MoCo-v2 (Chen et al., 2020b) | FPN | ResNet50 | 40k | 200 | 23M | $2 \times 224^2$ | 35.8 | 77.6 | 45.1 |
| SwAV (Caron et al., 2020) | FPN | ResNet50 | 40k | 200 | 23M | $2 \times 224^2$ | 35.4 | 77.5 | 44.9 |
| DenseCL (Wang et al., 2021) | FPN | ResNet50 | 40k | 200 | 23M | $2 \times 224^2$ | 37.2 | 78.5 | 47.1 |
| MocoV3 (Chen et al., 2021) | FPN | ViT-S/16 | 40k | 300 | 23M | $2 \times 224^2$ | 35.3 | 78.9 | 45.9 |
| MoBY (Xie et al., 2021b) | FPN | ViT-S/16 | 40k | 300 | 23M | $2 \times 224^2$ | 39.5 | 79.9 | 50.5 |
| DINO (Caron et al., 2021) | FPN | ViT-S/16 | 40k | 300 | 23M | $2 \times 224^2$ | 38.3 | 79.0 | 49.4 |
| DINO (Caron et al., 2021) | UperNet | ViT-S/16 | 160k | 300 | 23M | $2 \times 224^2$ | 42.3 | 80.4 | 52.7 |
| SelfPatch (Yun et al., 2022) | FPN | ViT-S/16 | 40k | 200 | 23M | $2 \times 224^2$ | 41.2 | 80.7 | 52.1 |
| SelfPatch (Yun et al., 2022) | UperNet | ViT-S/16 | 160k | 200 | 23M | $2 \times 224^2$ | 43.2 | 81.5 | 53.9 |
| DINO (Caron et al., 2021) | UperNet | ViT-S/16 | 160k | 800 | 23M | $2 \times 224^2 + 10 \times 96^2$ | 44.4 | 81.7 | 55.5 |
| iBOT (Zhou et al., 2022) | UperNet | ViT-S/16 | 160k | 200 | 23M | $2 \times 224^2$ | 44.1 | 81.4 | 55.3 |
| ADCLR (Zhang et al., 2023b) | UperNet | ViT-S/16 | 160k | 200 | 23M | $2 \times 224^2$ | 44.3 | 81.9 | 55.1 |
| PQCL (Zhang et al., 2023a) | UperNet | ViT-S/16 | 160k | 200 | 23M | $2 \times 224^2$ | 45.1 | 82.0 | 56.1 |
| CR2PQ (Ours) | UperNet | ViT-S/16 | 160k | 200 | 23M | $2 \times 224^2$ | **47.0** | **83.1** | **57.4** |

Table 3: Finetune top-1 and top-5 classification accuracies on ImageNet-1K with ViT-S and ViT-B. "PT Eps" and "FT Eps" mean the number of epochs for pretraining and fine-tuning, respectively.

| Method | Backbone | #Params | PT Eps | FT Eps | Top-1 | Top-5 | Backbone | #Params | PT Eps | FT Eps | Top-1 | Top-5 |
|---|---|---|---|---|---|---|---|---|---|---|---|---|
| DeiT (Touvron et al., 2021) | ViT-S | 22M | 300 | 200 | 79.9 | $\sim$ | ViT-B | 86M | 300 | 100 | 81.2 | $\sim$ |
| MAE (He et al., 2021) | ViT-S | 22M | 1600 | 200 | 80.6 | $\sim$ | ViT-B | 86M | 300 | 100 | 83.2 | $\sim$ |
| Moco V3 (He et al., 2021) | ViT-S | 22M | 1600 | 200 | 81.4 | $\sim$ | ViT-B | 86M | 800 | 100 | 83.2 | $\sim$ |
| DINO (He et al., 2021) | ViT-S | 22M | 1600 | 200 | 81.5 | $\sim$ | ViT-B | 86M | 800 | 100 | 82.8 | $\sim$ |
| iBOT (Zhou et al., 2022) | ViT-S | 22M | 800 | 200 | 81.8 | $\sim$ | ViT-B | 86M | 300 | 100 | 83.2 | $\sim$ |
| TinyMIM (Ozbulak et al., 2023) | ViT-S | 22M | 300 | 100 | 81.5 | 85.8 | ViT-B | 86M | 300 | 100 | 83.4 | 96.3 |
| CR2PQ (Ours) | ViT-S | 22M | 300 | 200 | **82.2** | **86.1** | ViT-B | 86M | 300 | 100 | **83.7** | **96.5** |

## 4.2 ABLATION STUDIES

**Fine-tuning classification.** Most of the previous dense contrastive learning methods (Yun et al., 2022; Xie et al., 2021c) show worse performance compared with global instance-level contrastive methods (Zhou et al., 2022; Caron et al., 2021), which is because the patch-level loss could inhibit the model from learning global-discriminative information. However, different from previous DCL methods, we find CR2PQ could achieve comparable results on fine-tuning classification with the instance-level SSL methods. We guess that's because we add the global objective in Eq. 10, which helps CR2PQ learn global-discriminative information. Table 3 shows the fine-tuning accuracies with different backbones (ViT-S and ViT-B). With ViT-S, CR2PQ achieves 82.2 top-1 accuracy with only 300 epochs pretraining, which outperforms previous SOTA iBOT (Zhou et al., 2022) with 800 epochs pretraining by 0.4 point. With ViT-B, our methods achieve 83.7 top-1 accuracies with only 300 epochs pretraining, which outperforms under the same epochs pretraining.

**Teacher models and architectures.** To explore the effect of the teacher model, we test our CR2PQ with different teacher models. We also evaluate our CR2PQ without teacher models (See EMA update and Pixel) in Table 4. Specifically, we adopt ViT-S (Dosovitskiy et al., 2020) as the student model and use ViT-B, ResNets (He et al., 2016), and ViT-L as the teacher models. As the output dimensions of

Table 4: Comparisons with different teachers on the COCO datasets (detection and segmentation).

| Student | Teacher | Epoch | mAP$^{bb}$ | mAP$^{mk}$ |
|---|---|---|---|---|
| ViT-S | EMA update (Contrastive) | 300 | 44.1 | 39.8 |
| | Pixel (Masked Image Modeling) | | 45.2 | 40.8 (+1.0) |
| | DINO (Caron et al., 2021) (Res-50) | | **47.4** | **41.8** (+2.0) |
| | DINO (Caron et al., 2021) (ViT-B) | | 47.0 | 41.1 (+1.3) |
| | SimCLR (Chen et al., 2020a) (Res-50) | | 47.1 | 41.6 (+1.8) |
| | MAE (He et al., 2021) (ViT-B) | | 46.5 | 40.7 (+0.9) |
| | MAE (He et al., 2021) (ViT-L) | | 47.0 | 41.4 (+1.6) |
| | iBOT (Zhou et al., 2022) (ViT-B) | | 46.2 | 41.0 (+1.2) |
| | iBOT (Zhou et al., 2022) (ViT-L) | | 46.8 | 41.5 (+1.7) |

different teacher models could be variant (*e.g.*, $14 \times 14 \times 768$ for ViT-B/16, $7 \times 7 \times 2048$ for ResNets), we change the grid size $K_B^H$ and $K_B^W$ of the relative positional index. More concretely, for ResNet-50 and ResNet-101, we set the grid size of the positional embeddings to $224/7 = 32$. For ViT-B/16, we set the grid size of the positional embeddings to $224/14 = 16$. We mainly choose the teacher model pre-trained by some profound methods, including MAE (He et al., 2021), iBOT (Zhou et al., 2022), and DINO (Caron et al., 2021). We keep the batch size as 2048 and use ViT-S/16 as the student network to train CR2PQ with 300 epochs. Table 4 shows the results with different teacher models. We find using contrastive-based teacher model (Caron et al., 2021; Chen et al., 2020a), our CR2PQ could obtain more gains than using MIM-based (He et al., 2021; Zhou et al., 2022) teacher models. We guess that's because contrastive-based methods usually learn global-discriminative information, but lose local information. When using these contrastive-based methods, our CR2PQ could enhance the ability to capture the local information. In contrast, MIM-based

Table 5: **Ablation on projector head**. We pre-train all the models with 400 epochs and 2048 batch size on ImageNet-1K. We compare different architectures of heads. "Share weights" means the global and local projectors share the same weights. "Trans" means multi-head self-attention decoder.

| Architecture | Share weights | Global projector | Local projector | Top-1 | Top-5 | Framework | mAP$^{bb}$ | mAP$^{mk}$ |
|---|---|---|---|---|---|---|---|---|
| ViT-B/16 | ✓ | 2-layer MLP | 2-layer MLP | 83.3 | 96.2 | Cascade-RCNN | 52.0 | 45.1 |
| | ✓ | 3-layer MLP | 3-layer MLP | 83.2 | 96.1 | | 52.4 | 45.3 |
| | ✗ | 2-layer MLP | 2-layer MLP | 83.4 | 96.2 | | 52.0 | 45.1 |
| | ✗ | 3-layer MLP | 2-layer MLP | 83.2 | 96.1 | | 52.1 | 45.0 |
| | ✗ | 2-layer MLP | 3-layer MLP | 83.5 | 96.4 | | 52.5 | 45.1 |
| | ✗ | 2-layer MLP | 2-layer Trans | **83.7** | **96.5** | | **53.0** | **45.6** |

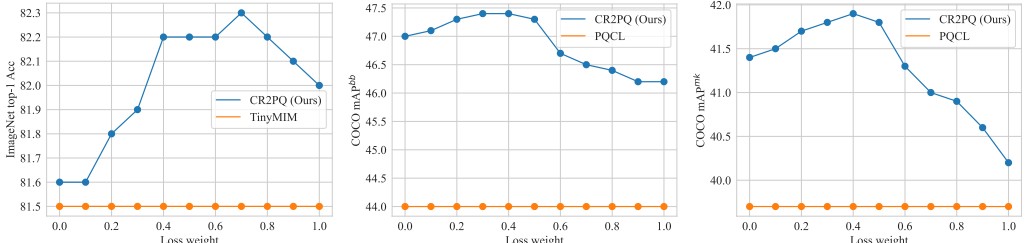

Figure 3: Finetuning classification on ImageNet-1K (left), object detection (middle) and instance segmentation (right) performance on COCO with different global and local objective weights $\lambda$. All the experimental settings keep the same for fair comparisons.

methods usually add the patch-level objective, and the pre-trained models are more spatial-sensitive but less global-discriminative. Therefore, when using MIM-based methods as the teacher model, our CR2PQ may lose some global-discriminative information, leading to fewer improvements than using contrastive-based teacher models. Besides, It's noteworthy that our CR2PQ w/o pretrained teacher (using pixels as the teacher) can still outperform previous SOTA method (Zhang et al., 2023a) with a large range, which further demonstrates the effectiveness of the continuous coordinate-based task.

**Loss weight of global semantic learning.** As CR2PQ mainly focuses on dense prediction tasks, we set the weight of the global objective $\lambda = 0.5$ as the default. To better balance the global-discriminative and spatial-sensitive information, we further conduct a set of experiments by switching $\lambda$ from $0 \sim 1$. Specifically, we pre-train ViT-S/16 300 epochs with 2048 batch size. Then, for detection and segmentaion, we finetune the pretrained model on COCO dataset with Mask RCNN framework. Fig. 3 illustrate the funetuning classification accuracy on ImageNet, mAP$bb$ and map$^{mk}$ results on COCO with different global loss weight $\lambda$. We find with larger $\lambda$, CR2PQ obtains higher fine-tune accuracy on ImageNet-1K. However, correspondingly, large $\lambda$ makes CR2PQ learn less spatial information, resulting mAP score drops on the COCO dataset. We also find when setting $\lambda = 0$, the detection score is also lower than $\lambda = 0.5$, we guess that's because too small $\lambda$ makes CR2PQ fail to learn global-discriminative information, where detection tasks also require the discriminative information to help classify the object in classification head layer.

**Projector head of global semantic learning.** To further study the effect of the local and global projector head, we try different combinations of the two modules. Following SimSiam (Chen & He, 2021) and Barlow Twins (Zbontar et al., 2021), we mainly attempt to use 2-layer and 3-layer MLP and 2-layer transformer blocks as projector heads. Since the global projector inputs one global token, we only add the transformer block on the local projector module. Table 5 shows fine-tuned classification, detection and instance segmentation results on ImageNet-1K and COCO datasets with different architectures. Different from previous contrastive methods (Chen & He, 2021; Zbontar et al., 2021), we find that lightweight global projectors can bring better classification accuracy than complex architectures, while employing two-layer transformer layer as local projector can bring more gains than simply using two-layer and three-layer MLP.

**Pretraining Strategy.** To show the effectiveness of our continuous rotary positional embedding, we conduct a set of experiments, including three types of positional embeddings, *i.e.*, continuous sin-cos embedding, discrete rotary positional embedding (RoPE), learnable relative positional embeddings and discrete sin-cos embeddings. More concretely, for sin-cos embedding, we first calculate the relative positional index proposed in Eq. 4. Then, for each index, we calculate the sin-cos embedding (He

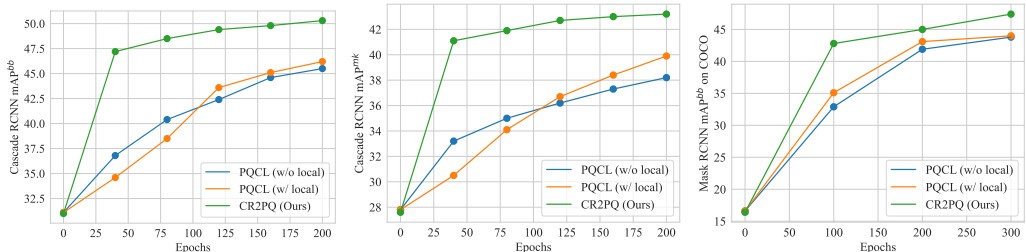

Figure 4: Finetuning Cascade-RCNN and Mask-RCNN (right) on detection (left) and segmentation (middle) on COCO dataset. We report the mAP$^{bb}$ and mAP$^{mk}$ performance for each 40 epochs.

et al., 2021) through the given relative indices of each patch. For discrete positional embeddings, we perform pretraining in MAE, since the discrete position can **not** represent the relative positions of two views. Therefore, for each image, we randomly mask the given image with 75% masking ratio. Then, we use the remain part to reconstruct the masked part. For learnable positional embedding, we first calculate the relative positional index proposed in Eq. 4. Then, we use a two-layer MLP to map the tuple $(h, w)$ to $D_{model}$ dimension as the positional embedding. For discrete sin-cos, we also adopt the MAE pretraining and the model degenerates to MAE (He et al., 2021) (replace the RGB supervision with teacher embeddings). Table 6 shows the classification and detection results on ImageNet and COCO, where our CR2PQ significantly outperforms peer methods.

**Convergence and training efficiency.** To explore the convergence of the proposed CR2PQ, we conduct a group of experiments with different pretraining epochs. Fig. 4 shows the detection and segmentation results on the COCO dataset with different pretraining epochs using Cascade-RCNN and Mask-RCNN frameworks. We surprisingly find with Cascade-RCNN, our CR2PQ can achieve 47.2 mAP$^{bb}$

Table 6: **Ablation on different pretraining strategy**. We pre-train ViT-S/16 with 300 epochs on ImageNet-1K and fine-tune on other datasets.

| Student | Positional embedding | Epoch | Acc | mAP$^{bb}$ |
|---|---|---|---|---|
| | Continuous 2D RoPE | | **82.2** | **50.5** |
| | Discrete RoPE | | 81.5 | 48.2 (-2.3) |
| ViT-S/16 | Continuous Sin-cos | 300 | 81.9 | 47.8 (-2.7) |
| | Learnable | | 81.3 | 46.9 (-3.6) |
| | Discrete Sin-cos | | 81.1 | 45.9 (-4.6) |

and 41.1 mAP$^{mk}$ score with only 40 epochs pretraining, which obtains **10.4** improvements on mAP$^{bb}$ and **7.9** improvements mAP$^{mk}$ scores on detection and segmentation tasks, respectively. Besides, we also find with local loss, PQCL converges slower than without using the local loss (when 40 epochs pretraining, PQCL (w/o local gets better results than w/ local)). However, with 200 epochs pretraining, PQCL (w/ local) outperforms PQCL (w/o local) by 0.7 mAP$^{bb}$ and 1.7 mAP$^{mk}$ scores. We guess that's because the patch-level loss could impede the model from learning global-discriminative information, which slows down the convergence rate. However, the patch-level loss also makes the model learn spatial-sensitive information, resulting in better performance on detection and segmentation under long epochs pretraining. In contrast, CR2PQ enjoys fast convergence, and it outperforms previous SOTA by a large range with only 40 epochs pretraining.

# 5   CONCLUSION

In this paper, we propose CR2PQ to learn dense representations through continuous relative rotary positional embedding. Different from prior 2D RoPE works, we extend the discrete RoPE to continuous and demonstrate its effectiveness in learning dense representations from comprehensive experiments. Then, we conduct exhaustive ablation studies to demonstrate the robustness of the proposed CR2PQ, where our method achieves new SOTA results on detection and segmentation tasks, outperforming previous dense contrastive learning SOTA with a large range. Besides, our CR2PQ also achieves comparable results on classification with previous methods. Finally, we demonstrate the effectiveness of the proposed continuous RoPE through different positional embedding types.

**Limitations.** While CR2PQ achieves state-of-the-art results, there are limitations to consider for future exploration: Sensitivity to Background-heavy Views: The current method relies on randomly cropped views for contrastive learning (or distillation). In some cases, these views might contain mostly background with little foreground content. This can lead the model to learn irrelevant information that hinders performance on various downstream tasks focused on the foreground objects.

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

## A    IMPLEMENTATION AND EXPERIMENT SETUPS

**Baselines.** We consider recent advanced self-supervised methods based on the ResNets (He et al., 2016) and ViTs (Dosovitskiy et al., 2020) architectures: (a) self-supervised ResNets: SimCLR (Chen et al., 2020a), MoCo-v2 (Chen et al., 2020b), SwAV (Caron et al., 2020), Barlow Twins (Zbontar et al., 2021), ZeroCL (Zhang et al., 2021), ARB (Zhang et al., 2022), DenseCL (Wang et al., 2021), ReSim (Xiao et al., 2021), and DetCo (Xie et al., 2021a); and (b) self-supervised ViTs: DINO (Caron et al., 2021), MoCo-v3 (Chen et al., 2021), MoBY (Xie et al., 2021b), iBOT (Zhou et al., 2022), SelfPatch (Yun et al., 2022), TinyMIM (Ozbulak et al., 2023).

**Code of computing relative coordinates.** Code 1 shows the implementation of how to compute relative coordinates matrix of $\mathbf{rp}_B$.

```
def factorial(p_A, p_B, K_A, K_B):
    # p_A and p_B are given in line 199 of this paper.
    # K_A, K_B are given in Eq.3 in this paper.
    # width
    w_per_grid_A = p_A[3] / K_A
    w_grid_bias = (p_B[1] - p_A[1]) / w_per_grid_A
    w_per_grid_B = p_B[3] / K_B
    w_grid_scale = w_per_grid_B / w_per_grid_A
    # height
    h_per_grid_A = p_A[2] / K_A
    h_grid_bias = (p_B[0] - p_A[0]) / h_per_grid_A
    h_per_grid_B = p_B[2] / K_B
    h_grid_scale = h_per_grid_B / h_per_grid_A
    # compute coordinate matrix
    h_start, h_end = h_grid_bias + K_B * h_grid_scale
    w_start, w_end = w_grid_bias + K_B * w_grid_scale
    grid_h = torch.arange(start=h_start, end=h_end, step=
        h_grid_scale)
    grid_w = torch.arange(start=w_start, end=w_end, step=
        w_grid_scale)
    grid = torch.meshgrid(grid_w, grid_h)
    return grid
```

Listing 1: Computing Relative Coordinates

## B    EXPERIMENT ON LARGE-SCALE MODEL

To explore the scalability of the proposed CR2PQ, we conduct experiments with ViT-L, and evaluate the pretrained ViT-Base on ViTDet (Li et al., 2022) detector. Specifically, we pre-train the ViT-Large with 800 epochs with batch size 2048, distributed on 16 A100 GPUs with the base learning rate 1.5e-4. Table B shows the results on ImageNet-1K using ViT-L backbone, where our CR2PQ can consistently outperform previous baselines on both 400 and 800 epochs pretraining. Table B shows the results of the COCO dataset when using ViT-B as the backbone and ViTDet as the detector head.

| Method | Architecture | Epoch | Acc@1 |
|---|---|---|---|
| DINO (Caron et al., 2021) | ResNet-50 | 400 | 77.4 |
| Moco V3 (Chen et al., 2021) | ViT-L/16 | 600 | 84.1 |
| MAE (He et al., 2021) | ViT-L/16 | 400 | 84.3 |
| CR2PQ (Teacher ResNet-50 DINO) | ViT-L/16 | 400 | **84.6** |
| MAE (He et al., 2021) | ViT-L/16 | 800 | 84.6 |
| iBOT (Zhou et al., 2022) | ViT-L/16 | 1000 | 84.8 |
| CR2PQ (Teacher ResNet-50 DINO) | ViT-L/16 | 800 | **85.3** |

Table 7: Finetuning classification results on ImageNet-1K dataset using ViT-L.

| Method | mAP$^{bb}$ | mAP$^{mk}$ |
|---|---|---|
| Scratch | 48.1 | 42.6 |
| MAE (He et al., 2021) | 51.1 | 45.6 |
| DINO (Caron et al., 2021) | 49.0 | 43.4 |
| CR2PQ (Ours) | **52.2** | **46.5** |

Table 8: Object detection and instance segmentation results on COCO datasets using ViT-Base and ViTDet.

