# OpenReview forum: "CR2PQ: Continuous Relative Rotary Positional Query for Dense Visual Representation Learning"
_ICLR.cc/2025/Conference — ICLR 2025 Poster_

### Official Review · Reviewer_MHTX · 2024-10-23

**Soundness:** 3
**Presentation:** 2
**Contribution:** 3
**Rating:** 6
**Confidence:** 3

**Summary:**

The paper presents a novel self-supervised framework for dense visual representation learning, which avoids the need for explicit dense correspondences between local features across views. Instead, the framework reframes the task as predicting local representations from one view to another, guided by relative positional cues. It integrates rotary positional embeddings within the student model and distills knowledge from a pre-trained, frozen teacher model. This approach yields faster convergence and improved performance on standard benchmark evaluations.

**Strengths:**

- The proposed self-supervised framework for dense visual representation learning is novel.
- The method elegantly eliminates the need to establish explicit correspondence between local features across views by leveraging relative positional cues.
- The performance on dense downstream tasks is thoroughly evaluated, showing faster convergence and achieving state-of-the-art results on standard benchmarks.

**Weaknesses:**

- The method differs from existing baselines in three key ways: (1) the use of rotary positional embeddings, (2) the use of a pre-trained, frozen teacher model, and (3) the proposed pretext task. This makes it challenging to assess the contribution of each component to the overall performance. Specifically, the fairness of the experimental setup is questionable, as other methods are trained from scratch while CR2PQ benefits from a pre-trained teacher. More ablation studies are needed to separate the impact of each element.

- Overall, the writing is difficult to follow, with multiple notation inconsistencies, typos, and signs of negative vertical spacing used to fit within the page limit.

- Equation 1 is misleading/incorrect as it suggests that the representation of a single patch is independent of its context.
- Equation 2: The angle of the key seems incorrect.
- Line 210: The image dimensions are inconsistent with line 157.
- Line 214: Inconsistent use of $\mathbf{p}{a}$ and $\mathbf{p}{A}$.
- Line 234: The notation is inconsistent with the left side of Equation 3.
- Table 1: Framwork $\rightarrow$ framework.
- Figure 1: There seem to be inconsistencies in the notations used within the figure and also with respect to the method section.
- "pertaining" $\rightarrow$ "pretraining"/"pre-training" (11 occurrences).
- Line 86: exhausted $\rightarrow$ exhaustive.
- Line 161: $\mathbf{W}{pos}$ $\rightarrow$ $\mathbf{P}^{i}{pos}$.

**Questions:**

- In Table 1, what does the row "RoPE" exactly correspond to? A ViT-S/16 equipped with rotary positional embedding, randomly initialized and finetuned on the downstream task?

- In Table 4, what does the row "EMA update (Contrastive)" exactly correspond to? Is the teacher randomly initialized?

- At line 219. it is mentioned that the patch size of view A is set to 1, but then it is set to $p_{A}$. Can you clarify this?

- At line 227: I suggest using another notation for $p_{A}$ as the patch size, as it is confusing.

---

> ### Author Response · Authors · 2024-11-20
> **Response to Reviewer MHTX**
>
> Thank you for the time, thorough comments, and nice suggestions. In the following response, we answer your comments/questions point-by-point, clarify the effectiveness of the proposed modules, and supplement new experiments as suggested to further strengthen our contributions.
>
> > Q1. Ablation studies on different components.
>
> - **The use of rotary positional embedding**. This ablation study is given in Table 6 in our paper. Here we emphasize the results:
>
> | Position encoding | Epoch | Acc | mAP$^{bb}$ |
> | ------ | ----- | --- | ---------- |
> | RoPE   | 300   | **82.2** | **50.5** |
> | Learnable | 300 | 81.3 | 46.9 |
> | Sin-Cos | 300 | 81.9 | 47.8 |
>
> The results demonstrate the effectiveness of RoPE, especially on the detection task (mAP$^{bb}$), as the RoPE exhibits better extrapolation capability when feeding high-resolution images in the detection task.
>
> - **The use of pre-trained teacher model**. This ablation is given in Table 4 in our paper in our revised version. Here we emphasize the results:
>
> | Method | Teacher | mAPbb | mAPmk |
> | ------ | ------- | ----- | ----- |
> | CR2PQ (Ours) | None (Raw Pixel) | **45.2** | **40.8** |
> | PQCL | None | 44.0 | 39.7 |
> | DINO | None | 40.8 | 37.3 |
>
> The results demonstrate our model (w/o teacher model) still outperforms previous SOTA PQCL with a large range.
>
> - **The proposed pretext task**. The results of this ablation study are given in Table 6 in our revised version. Here are the results:
>
> | Position encoding | Epoch | Acc | mAP$^{bb}$ |
> | ------ | ----- | --- | ---------- |
> | Cross View (Ours) + Continuous RoPE (Ours)   | 300   | **82.2** | **50.5** |
> | Single View + Discrete RoPE | 300 | 81.5 | 48.2 |
>
> Single View + Discrete RoPE means directly applying RoPE in MAE. The results show the effectiveness of the proposed pretext tasks.
>
> > Q2. Typos and Writings.
>
> - Thank you for your careful reading of our paper and for pointing out these typos and confusions, which have given us the opportunity to correct them. We have modified these typos, rewritten the relative coordinates in our revised version, and highlighted these changes in blue font. Besides, we made an illustration of how to compute the relative coordinates in Figure 2 in our revised version. Please refer to our revised paper.
>
> > Q3. In Table 1, what does the row "RoPE" exactly correspond to?
>
> - Yes, it means directly using the RoPE in the ViT-S, and randomly initializing and finetuning on downstream tasks.
>
> > Q4. What does the row "EMA update (Contrastive)" exactly correspond to?
>
> - EMA update means we follow the iBOT and PQCL, randomly initializing the teacher, and use EMA to update the parameters of the teacher model.
>
> > Q5. It is mentioned that the patch size of view A is set to 1.
>
> - Thanks for pointing out this confusion. The patch size is $p_A$ and after patching, each token will take a position, where the interval is set to 1. We have revised it to "the scale of the coordinates system is 1." for better understanding. Please refer to line 213, Eq. 3, and Fig. 2 in our revised version.
>
> > Q6. Using another notation for $p_A$.
>
> - Thanks for your constructive suggestions. We have replaced the patch size $p_A$ with $ps_A$ to better differentiate from the absolute position $\mathbf{p}_{A}$.
>
> Thanks again for the valuable suggestions, and please let us know if you have any further questions.

---

> > ### Comment · Reviewer_MHTX · 2024-11-20
> > **Response to authors**
> >
> > Thank you for your answer.
> > Some points of your answer still confuse me.
> > I couldn't find the exact description of Table 4 or the table corresponding to the ablation on the teacher model in your answer?
> > Do these tables correspond to a finetuning on MS-COCO with Mask-RCNN?
> >
> > Thank you in advance for the clarification.

---

> > > ### Author Response · Authors · 2024-11-20
> > > **Response to Reviewer MHTX**
> > >
> > > Thank you for your response. Here are our clarifications for your confusion.
> > >
> > > > w/ and w/o teacher model.
> > >
> > > - The **first two raws** (EMA and Pixel) in Table 4 show the results **w/o the pretrained teachers**. EMA means the teacher is randomly initialized and using the EMA to update its parameters. Pixel means directly using the pixel values of view $B$ as the target.
> > >
> > > > Do these tables correspond to a finetuning on MS-COCO with Mask-RCNN?
> > >
> > > - Yes. Acc means fine-tuning on ImageNet-1K, while mAP$^{bb}$ and mAP$^{mk}$ mean the results on the COCO dataset.
> > >
> > > Please let us know if you have any further questions!

---

> > > > ### Comment · Reviewer_MHTX · 2024-11-20
> > > > **Response to clarification**
> > > >
> > > > Thank you for the clarification.
> > > >
> > > > I recommend providing a clearer description of Table 4 (finetuning method and dataset) in the paragraph titled "Teacher models and architectures."
> > > >
> > > > Overall, I believe the merits of this draft outweigh its shortcomings, and I am willing to raise my score.

---

> > > > > ### Author Response · Authors · 2024-11-20
> > > > > **Response to Reviewer MHTX**
> > > > >
> > > > > Thank you for your further constructive suggestions.
> > > > >
> > > > > We have added the description including (the meaning of EMA and Pixels, used method and datasets) to the "Teacher models and architectures" parts in our revised version.
> > > > >
> > > > > Please let us know if you have any further questions or suggestions!

---

### Official Review · Reviewer_iV1q · 2024-10-30

**Soundness:** 2
**Presentation:** 3
**Contribution:** 3
**Rating:** 6
**Confidence:** 2

**Summary:**

1. The paper introduces Continuous Relative Rotary Positional Query (CR2PQ), a novel method for dense visual representation learning.
CR2PQ addresses the challenge of establishing pixel/patch correspondence across different views in dense contrastive learning (DRL) by transforming discrete positional embeddings to continuous representations.

2. It utilizes a rotary positional embedding to represent the relative positions between two views and reconstructs the latent representations of one view from another through a rotary positional query.

3. The method simplifies the dense contrastive learning paradigm by making it correspondence-free and integrates easily into various representation learning frameworks.

4. Extensive experiments on standard datasets demonstrate state-of-the-art (SOTA) results, outperforming the previous SOTA method (PQCL) significantly in detection and segmentation tasks on COCO with improved mAP scores.

**Strengths:**

1. CR2PQ introduces a pioneering method for dense visual representation learning by utilizing continuous relative rotary positional embeddings, which is a significant departure from traditional discrete embeddings.

2. The method achieves state-of-the-art results across various benchmarks, including object detection and segmentation tasks on COCO and semantic segmentation on ADE20K, outperforming previous leading methods by a considerable margin.

3. The introduction of a positional-aware cross attention module enhances the learning of semantic information without incurring significant additional computational costs. CR2PQ's use of rotary positional embeddings makes it robust to various view augmentations, including random cropping, which is a common challenge in contrastive learning methods.

4. The paper supports the method's strengths through extensive experiments and ablation studies, providing a thorough analysis of CR2PQ's performance under different conditions and configurations.

**Weaknesses:**

1. Experiments. The author should provide more scales of backbone to validate the scalability of the method. Most experiments are conducted on ViT-S. The reviewer understands the efficiency of the experiments, however, there should be some experiments on larger backbones.

**Questions:**

1. What is the performance of the CR2PQ backbone performance on some strong detectors, such as DINO or Co-DETR?

2. CR2PQ requires the teacher model to provide contrastive pairs, however, the performance does not improve as the model becomes larger (ViT-L vs ResNet50). The reviewer wonders about the performance of a larger model for the student. Does this approach work for a larger backbone as a student, such as ViT-L/ViT-G? The authors are suggested to validate the scalability of the method.

3. Some small mistakes

- The font of the paper is different from other papers. Should it be correct?

- line 274, there is an overlap between the table and the caption.

---

> ### Author Response · Authors · 2024-11-20
> **Response to Reviewer iV1q**
>
> Thank you for the time, thorough comments, and nice suggestions. We supplement new experiments as suggested to further address your concerns about scalability.
>
> > Q1. Scalability of CR2PQ
>
> - We have conducted the experiment on ViT-L, using ResNet-50 pre trained with DINO as the teacher. We pre-train the model with 800 epochs with batch size 2048, distributed on 16 A100 GPUs with the base learning rate 1.5e-4. We evaluate the pre-trained model on the finetuning classification task. Here are the additional results:
>
> | Method | Architecture | Epoch | Acc@1 |
> | ----- | ----- | ---- | ---- |
> | DINO | ResNet-50 | 400 | 77.4 |
> | Moco V3 | ViT-L/16 | 600 | 84.1 |
> | MAE  | ViT-L/16  | 400 | 84.3 |
> | CR2PQ (Teacher ResNet-50 DINO) | ViT-L/16 | 400 | **84.6** |
> | MAE  | ViT-L/16  | 800 | 84.6 |
> | iBOT | ViT-L/16  | 1000 | 84.8 |
> | CR2PQ (Teacher ResNet-50 DINO) | ViT-L/16 | 800 | **85.3** |
>
> Our methods can stably outperform previous contrastive (Moco V3) and masked image modeling (MAE, iBOT) methods. Although the phenomenon of using a smaller teacher to distill a larger student brings gains is an interesting one, it is also necessary to acknowledge that our method, even without a teacher and using pixels as the training objective, can achieve state-of-the-art results (see Table 4 in our revised version).
>
> > Q2. DINO or Co-DETR
>
> We feel sorry we may not provide results with DINO or Co-DETR head, as we do not find previous SSL baselines using the two detector heads (We don't have enough time to evaluate the previous baselines on these two methods.). We alternatively provide the results using ViTDet detector [1] with ViT-B/16, and here are the results on the COCO dataset:
>
> | Method | mAP$^{bb}$ | mAP$^{mk}$ |
> | ------ | ---------- | ---------- |
> | Scratch| 48.1 | 42.6 |
> |MAE | 51.1 | 45.6 |
> |DINO | 49.0 | 43.4 |
> | CR2PQ | **52.2** | **46.5** |
>
> Our methods can stably outperform previous contrastive and masked image modeling methods. We hope the results of the ViTDet can alleviate your concerns.
>
> > Q3. Typos.
> - We have modified the template and the overlap. Please check our revised version.
>
> [1] Li Y, Mao H, Girshick R, et al. Exploring plain vision transformer backbones for object detection[C]//ECCV 2022.
>
> Thanks again for the valuable suggestions, and please let us know if you have any further questions.

---

> ### Comment · Reviewer_iV1q · 2024-11-20
> **Response to authors**
>
> Thanks for your response. After reading your rebuttal, most of my concerns have been well addressed. Therefore, I decide to raise my score to borderline accept :) Besides, I strongly suggest the author put these experiments in the main context of the manuscripts, maybe in the appendix.

---

> > ### Author Response · Authors · 2024-11-20
> > **Response to Reviewer iV1q**
> >
> > Thank you for your response and further suggestions.
> >
> > We now have added these results to Appendix B in our revised version.
> >
> > Please let us know if you have any further questions or suggestions!

---

### Official Review · Reviewer_p4vE · 2024-11-02

**Soundness:** 3
**Presentation:** 2
**Contribution:** 2
**Rating:** 6
**Confidence:** 3

**Summary:**

The authors propose a distillation technique where a student is densely trained to match teacher features. The novelty comes from using 2D RoPE in the network as well as a cross-attention module with relative positional information. They show good empirical results on detection and segmentation.

**Strengths:**

-The empirical results are good and outperform previous SOTA.

-I think this paper can be worthwhile to accept, I'm willing to improve my score based on the author's reply.

**Weaknesses:**

-L142: Relative positional encoding = RoPE?

-L161: W_{pos} v.s. P_{pos} ?

-The notation in equation 1 is confusing. It is as if the patches don’t interact with each other. I would use a new variable to define a patch representation. Also if f_\theta denotes the ViT, why does it take z as input, which already contains the linear layer on the left side of the equation but not on the right side. I think the notation should be made more precise.

-Equation 2 has some n and m mixed.

-L219: “we set each patch size of the view A as 1”, but in L227 p_A (the patch size) is defined?

-L228: There is a sentence “Since we set each grid size of the anchor view as 1.” What is that supposed to mean?

-L297: If I’m not mistaken, the definition of q doesn’t make sense.

-The first stated contribution is using 2D RoPE for SSL based methods. Then, in L358, shoud state “We also evaluate the detection and segmentation without pretraining i.e. directly using 2D RoPE”. First, that entry is only in Table 1 and not Table 2. Second, I think you should also independently show empirical evidence of your 2 first contributions (2D RoPE and cross-attention module) and report results for that.

-In general, I think the paper could be more explicitaly precise with how sizes/positions are encoded e.g. is it relative to the original image input grid or relative to the crop?



Minor:

-L082: “as the downstream task only input”

-If I’m not mistaken, there is a problem with sentence at L203 starting with i.e.

**Questions:**

-Why use a pretraining network for the teacher? You are comparing with other baselines which some of which learn everything from scratch. This seems like a logical thing to try, have you tried that?

---

> ### Author Response · Authors · 2024-11-20
> **Response to Reviewer p4vE**
>
> Thank you for the time, thorough comments, and nice suggestions. We are pleased to clarify your questions step-by-step.
>
> > Q1. Relative positional encoding = RoPE?
>
> - No. In this paper relative coordinates matrix $\mathbf{rp}\_{B}$ + RoPE = relative positional embedding. RoPE is a category of positional encoding, and it takes the index (or coordinate) of the position in the sequence as input and returns the positional embedding. The proposed relative coordinate matrix $\mathbf{rp}\_{B}$ can provide an accurate relative location between the two views, while the RoPE takes the $\mathbf{rp}_{B}$ as input and returns the continuous relative positional embeddings.
>
> > Q2. $\mathbf{W}\_{pos}$ v.s. $\mathbf{P}\_{pos}^{i}$
>
> - We have modift the $\mathbf{W}\_{pos}$ to $\mathbf{P}\_{pos}^{i}$ in our revised version.
>
> > Q3. The notation of Eq. 1 is confusing.
>
> - Many thanks for pointing out this mistake, and following your suggestion, we have modified it with a new variable $\mathbf{o}$. Please check Eq. 1 in our revised version.
>
> > Q4. Some $n$ and $m$ are mixed.
>
> - We have corrected it. Please check it in Eq. 2 in our revised version.
>
> > Q5. line 219 and line 228, patch size as 1.
>
> - Sorry for the confusion. We replace the original "relative positional index matrix" with "coordinates" for better understanding. Please check Line 214 in our revised version.
>
> > Q6. If I’m not mistaken, the definition of q doesn’t make sense.
>
> - Thanks for pointing out. We have replace the $\mathbf{q}^{(m,n)}$ with $\mathbf{rp}_{B}^{(m,n)}$. Please check line 260 in our revised version.
>
> > Q7. The first stated contribution is using 2D RoPE for SSL-based methods...
>
> - Actually, the first stated contribution is **continuous** RoPE for SSL methods. There are also some works attempting RoPE in image and video diffusion models. However, all of them directly integrate the discrete RoPE (the index of the patch is discrete). This paper focuses on modeling the positional relation between the two random cropped two views, where this means when we use the location of one view to represent the location of the other view, the relative coordinates matrix $\mathbf{rp}\_{B}$ of the other view will be continuous.
>
> > Q8. independently show empirical evidence of your 2 first contributions
>
> - Thanks for your nice suggestions. Here, we report the results on the ADE20K dataset of the two modules, which is given in Table 4 in our revised version.
>
> | Method | mIoU | aAcc | mAcc | Second Per Iteration |
> | -----  | ---- | --- | ---- | ---- |
> | CrossAttn + Continuous RoPE (Ours) | **47.0** | **83.1** | **57.4** | 1.24 sec |
> | SelfAttn + RoPE | 46.5 | 82.4 | 56.6 | 1.36 sec |
> | CrossAttn + SinCos | 45.3 | 82.1 | 56.3 | 1.24 sec |
>
> The new results demonstrate the effectiveness of the continuous RoPE and the efficiency of the proposed Cross Attention module.
>
> > Q9. Is it relative to the original image input grid or relative to the crop?
>
> - The relative coordinates of the two views depend on the location of the random crop.
>
> > Q10. typos
>
> We have corrected it and carefully checked the remaining parts of the paper again. Please check our revised version.
>
> > Q11. Why use a pretraining network for the teacher?
>
> - Some previous baselines (TinyMIM, SelfPatch) use a large pre-trained teacher model, while some baselines (PQCL, ADCLR) train from scratch. Therefore, we report both w/ and w/o teacher (use the pixel as the teacher) in Table 4 in our revised version. Here are the results on the COCO dataset under 300 epochs pretraining:
>
> | Method | mAP$^{bb}$ | mAP$^{mk}$ |
> | --------- | -------------- | -------------- |
> | CR2PQ w/ teacher | 47.4 | 41.8 |
> | CR2PQ w/o teacher | 45.2 | 40.8 |
> | PQCL (prev SOTA) | 44.0 | 39.7 |
>
> Our method w/o teacher (use pixels as the teacher) can still outperform previous SOTA by 1.2 mAP$^{bb}$ and 1.1mAP$^{mk}$ on the COCO dataset.
>
> Thanks again for the valuable suggestions, and please let us know if you have any further questions.

---

> > ### Author Response · Authors · 2024-11-22
> > **Look forward to your further reply**
> >
> > Dear Reviewer p4vE
> >
> > Approaching the ending of the discussion phase, we wonder whether our response and additional results address your concerns and whether you have further questions about our revised version.

---

> > > ### Comment · Reviewer_p4vE · 2024-11-25
> > > **Reply**
> > >
> > > Dear authors,
> > >
> > > Thank you for the detailed response.
> > >
> > > Regarding Q1, it was a rethorical question pointing to L142 (before the update) with what seemed like a wrong definition, sorry if it was not clear. Please correct this issue if it was the case.
> > >
> > > Most of my concerns have been adressed and I have raised my rating to lean towards acceptance.

---

> > > > ### Author Response · Authors · 2024-11-26
> > > > **Thanks for the reply**
> > > >
> > > > Thanks for your reply and further nice suggestions.
> > > >
> > > > We have modified it with "discrete relative positional embeddings". Meanwhile, we add the sentence "take the discrete patch index as input, and return the positional embeddings)" in our revised version for better understanding.
> > > >
> > > >
> > > > Please let us know if you have any further questions or suggestions.

---

### Official Review · Reviewer_qBT6 · 2024-11-04

**Soundness:** 3
**Presentation:** 2
**Contribution:** 2
**Rating:** 6
**Confidence:** 4

**Summary:**

The paper introduces the Continuous Relative Rotary Positional Query to enhance dense visual contrastive learning by improving pixel/patch correspondence across different views. It addresses limitations in existing self-contrasting methods by transforming discrete positional embeddings into continuous representations. The proposed CR2PQ enables more effective patch-level representation learning, achieving state-of-the-art results and faster convergence in detection and segmentation tasks on the COCO dataset.

**Strengths:**

1. Writing quality is good. The paper is well-structured, and clearly written.
2. SOTA performance. The paper demonstrates the state-of-the-art performance on mainstream detection and segmentation datasets, such as COCO and ADE20K, which is impressive.
3. Versatility of the method. The paper shows the simplicity of CR2PQ, which can be easily integrated into a variety of popular representation learning frameworks, such as mask-based learning, contrastive learning, and distillation methods.

**Weaknesses:**

1. Reliance on random cropping. Although random cropping can increase the variability of the input, its results may still be limited by the randomness of the cropping. In extreme cases, it may result in almost no overlap between the generated views, affecting the learning effect of the model.
2. Computational complexity. Complex matrix operations are required when calculating relative position embedding and rotating embedding, which increases the burden in scenarios with limited computing power.

P.S. There is an error in Figure 1. [CLS] should be global information, while patch is local information.

**Questions:**

Please refer to the weaknesses above.

---

> ### Author Response · Authors · 2024-11-20
> **Response to Reviewer qBT6**
>
> Thank you for the valuable and constructive suggestions. We appreciate the insightful comments from the reviewers, which have helped us to further refine and improve our work.
>
> > Q1. Reliance on random cropping.
>
> - Actually, our CR2PQ works well even for non-overlapping of the two views (view $A$ and view $B$), since we use the relative coordinate system to represent the different locations of the two views. We make a detailed illustration of the mechanism of computing relative coordinates in the **Figure 2** in our revised version to clarify your confusion.
>
> > Q2. Computational complexity of the RPE.
>
> - This operation can be easily done through the broadcast operation in Pytorch, which can be efficiently calculated. We give the Pytorch-like code of computing the relative index matrix $\mathbf{rp}\_B$ in Code 1 in the Appendix in our revised version. Here we also report the wall-clock time for one-iteration with 128 samples on one A100 GPU when computing the relative coordinates matrix $\mathbf{rp}_{B}$ to address your concern.
>
> | One iter | Computing $\mathbf{rp}_{B}$ | Model forward |
> | -------- | --------------------------- | ------------- |
> | 1.24 sec | 0.000127 sec (0.01%) | 1.1 sec (88.7%) |
>
> The table above shows that the time taken to compute the relative positional encoding (RPE) is less than **0.1%** of the time taken for a single forward pass of the model, which is almost **negligible**. The main reason for this is that we utilized PyTorch's broadcasting mechanism, thereby avoiding the use of loop statements and significantly improving computational efficiency.
>
> > Q3. [CLS] should be global information, while patch is local information.
> - Thanks for your nice comment. We have modified it in our revised version.
>
> Thanks again for the valuable suggestions, and please let us know if you have any further questions.

---

> > ### Comment · Reviewer_qBT6 · 2024-11-27
> >
> > Thanks for your reply. My concerns have been generally resolved. For Figure 1, the losses still need to be corrected to correspond. Please pay attention to these details to avoid confusing the readers.

---

> > > ### Author Response · Authors · 2024-11-27
> > > **Response to Reviewer qBT6**
> > >
> > > Thanks for your further suggestions.
> > >
> > > We have corrected the two losses in Figure 1 in our newly revised versions!

---

### Meta-Review · Area_Chair_9A98 · 2024-12-17

**Metareview:**

This paper presents a contrastive learning algorithm for visual pre-training. The algorithm is established upon dense contrastive learning (DCL) and adds a so-called Continuous Relative Rotary Positional Query (CR2PQ) to enhance patch-level learning. Experiments show improvement in standard visual pre-training benchmarks. After the rebuttal, the reviewers arrived at a consensus of weak acceptance. The AC finds no reason to overturn the reviewers' recommendation.

However, the AC is concerned about the timeliness of this research. The work was built upon dense contrastive learning and its performance is inferior to masked image modeling, the SOTA visual pre-training method. Note that the baseline (DCL) is a CVPR'21 paper that was first released on arXiv FOUR years ago. I am wondering how many researchers are still interested in contrastive learning (or how many new systems are built upon unlabeled contrastive learning) and, more importantly, whether the proposed positional query works in masked image modeling, which will be an important question regarding how much value the paper has provided to the community. While the AC suggests acceptance, **the authors are strongly required to add discussion about this topic in the final version**.

**Additional Comments On Reviewer Discussion:**

The paper initially got a mixed scores of 5 and 6, but after the rebuttal, all reviewers recommended 6, weak accpetance.

---

### Decision · Program_Chairs · 2025-01-22

Accept (Poster)